# Prussian Blue Nanozymes with Enhanced Catalytic Activity: Size Tuning and Application in ELISA-like Immunoassay

**DOI:** 10.3390/nano12101630

**Published:** 2022-05-10

**Authors:** Pavel Khramtsov, Maria Kropaneva, Artem Minin, Maria Bochkova, Valeria Timganova, Andrey Maximov, Alexey Puzik, Svetlana Zamorina, Mikhail Rayev

**Affiliations:** 1Faculty of Biology, Perm State University, 614068 Perm, Russia; kropanevamasha@gmail.com (M.K.); krasnykh-m@mail.ru (M.B.); mantissa7@mail.ru (S.Z.); mraev@iegm.ru (M.R.); 2Lab of Ecological Immunology, Institute of Ecology and Genetics of Microorganisms, 614081 Perm, Russia; timganovavp@gmail.com; 3Lab of Applied Magnetism, M.N. Mikheev Institute of Metal Physics of the UB RAS, 620108 Yekaterinburg, Russia; calamatica@gmail.com; 4Faculty of Biology and Fundamental Medicine, Ural Federal University Named after The First President of Russia B.N. Yeltsin, 620002 Yekaterinburg, Russia; 5Department of Analytical Chemistry and Expertise, Faculty of Chemistry, Perm State University, 614068 Perm, Russia; htb03starosta@gmail.com; 6Department of Mineralogy and Petrography, Faculty of Geology, Perm State University, 614068 Perm, Russia; alex.puzik@mail.ru; 7Core Facilities and Lab of Hydrochemical Analysis, Perm State University, 614068 Perm, Russia; 8Lab of Technological Mineralogy, Institute of Natural Science, Perm State University, 614068 Perm, Russia; 9Lab of Biogeochemistry of Technogenic Landscapes, Perm State University, 614068 Perm, Russia

**Keywords:** nanozyme, prussian blue, immunoassay, conjugate, peroxidase, gelatin, streptavidin, antibody, prostate-specific antigen, tetanus

## Abstract

Prussian blue nanozymes possessing peroxidase-like activity gather significant attention as alternatives to natural enzymes in therapy, biosensing, and environmental remediation. Recently, Prussian blue nanoparticles with enhanced catalytic activity prepared by reduction of FeCl_3_/K_3_[Fe(CN)_6_] mixture have been reported. These nanoparticles were denoted as ‘artificial peroxidase’ nanozymes. Our study provides insights into the process of their synthesis. We studied how the size of nanozymes and synthesis yield can be controlled via adjustment of the synthesis conditions. Based on these results, we developed a reproducible and scalable method for the preparation of ‘artificial peroxidase’ with tunable sizes and enhanced catalytic activity. Nanozymes modified with gelatin shell and functionalized with affine molecules were applied as labels in colorimetric immunoassays of prostate-specific antigen and tetanus antibodies, enabling detection of these analytes in the range of clinically relevant concentrations. Protein coating provides excellent colloidal stability of nanozymes in physiological conditions and stability upon long-term storage.

## 1. Introduction

Nanozymes are artificial nanomaterial-based catalysts, capable of mimicking the functions of enzymes [1,2]. Physical-chemical stability, low cost, and tunable properties make them a promising alternative to enzymes such as horseradish peroxidase (HRP), alkaline phosphatase, β-galactosidase that are used as labels in colorimetric immunoassays [3]. Colorimetric immunoassays include enzyme-linked immunosorbent assay (ELISA), lateral flow immunoassay (LFIA), dot-immunobinding assay, flow-through assay, and other less common modifications. In this type of assay, enzymes and their artificial analogs are responsible for the conversion of colorless substrate to colored product [4]. Nowadays, colorimetric immunoassays are one of the key analytical instruments in medicine, food quality control, and biotechnology [5,6]. Despite nanozymes not being mature enough and numerous challenges present [6], they emerge as prospective alternatives to natural enzymes in the field of biosensing [7,8].

In 2018, the Karyakin group reported a method for the preparation of Prussian blue nanoparticles with increased peroxidase-like activity denoted as ‘artificial peroxidase’ nanoparticles [9]. The key feature of ‘artificial peroxidase’ is the synthesis procedure based on the reduction of FeCl_3_/K_3_[Fe(CN)_6_] mixture with hydrogen peroxide (other reductants were also effective), whereas typically Prussian blue nanozymes are prepared by reaction between FeCl_3_ and K_4_[Fe(CN)_6_] [10,11] or by solvothermal method from the single precursor (e.g., K_3_[Fe(CN)_6_]) [12]. ‘Artificial peroxidase’ possesses almost tenfold higher catalytic activity in terms of K_cat_ in comparison with Prussian blue nanozymes prepared by the traditional approach. Moreover, the authors demonstrated that the increase of catalytic activity accompanied the growth of nanoparticle diameter, which was explained by the diffusion of a substrate to the inner parts of the nanoparticle body. Therefore the idea to synthesize large (200–300 nm) nanoparticles, functionalize them with recognition molecules, and use them as peroxidase-like labels in colorimetric immunoassay was very attractive. However, in the course of the preliminary experiments, we failed to reproduce the size control method described in the original paper [9], which relies on a change of ratio between iron salts and hydrogen peroxide (a related experiment and a possible explanation is given in Section 3.1). Prompted by the excellent catalytic properties of ‘artificial peroxidase’ which were harnessed for the creation of several sensitive assays [13,14,15], we decided to develop a more robust method for the preparation of ‘artificial peroxidase’ nanozymes with the desired size.

Prussian blue nanoparticles can be synthesized via different routes (extensively reviewed in the literature [16,17]). The most popular one is the mixing of FeCl_3_ and K_4_[Fe(CN)_6_], usually in the presence of citric acid which forms complexes with ferric ions and provides more homogeneous nanoparticles by decreasing the rate of nucleation [18]. Other methods include hydrothermal decomposition of a single precursor (e.g., K_3_[Fe(CN)_6_]) [19] or reduction of FeCl_3_/K_3_[Fe(CN)_6_] with H_2_O_2_ [20]. The latter approach enables the preparation of ‘artificial peroxidase’ nanozymes. The reductive approach is used infrequently; therefore, there is a limited number of studies dedicated to regulating the sizes of nanoparticles prepared by this method. In particular, Liu et al. prepared small (ca. 50 nm) nanoparticles by the gradual addition of FeCl_3_ to K_3_[Fe(CN)_6_] (both solutions contained H_2_O_2_) [20]. Sonication allows for further decreasing nanoparticle size to 5 nm [21]. Shiba et al. tuned the diameter of nanoparticles by varying the concentration of reductant (ascorbic acid) and addition of pre-formed nanoparticles in the reaction mixture (seed-growth method), however, the size of nanoparticles did not exceed 150 nm [22]. The utilization of polymers as capping agents is another option to regulate the size of Prussian blue nanoparticles [23,24,25].

Thus, this study aimed to develop a reproducible and scalable synthesis method for obtaining ’artificial peroxidase’ nanoparticles of various sizes and utilize them as labels in colorimetric immunoassays. We revealed that the addition of citric and oxalic acid enables precise tuning of nanoparticle size. Moreover, nanoparticles prepared in the presence of these acids possess higher catalytic activity in comparison with nanoparticles synthesized by the original method. Nanozymes with the highest peroxidatic activity were successfully applied for quantitative detection of the tumor marker prostate-specific antigen and human anti-tetanus antibodies.

## 2. Materials and Methods

Reagents and instrumentation used in this work can be found in the Appendix A.

### 2.1. Regulation of Size of ‘Artificial Peroxidase’ Nanoparticles by Changing the Concentration of Iron Salts and H_2_O_2_

The method of nanoparticle size regulation was based on the reported methodology [9]. Synthesis was performed in 10 mL glass vials at room temperature. The length of the magnet matched the diameter of the vial bottom. Then, 8.8 M hydrogen peroxide was quickly added (to 2.27–176 mM) into an equimolar aqueous solution of FeCl_3_ and K_3_[Fe(CN)_6_] (1.56–25 mM) under vigorous stirring (1000× *g*). The total volume of the mixture was 8.9 mL. After 60 min, the suspension was divided into 8 parts of 1 mL, transferred into centrifuge tubes, and centrifuged at 20,000× *g* for 60 min. The dark blue precipitate was washed three times with 1 mL of deionized water by centrifugation at 20,000× *g* for 15 min. After the final centrifugation, 100 μL of water was added to each tube. Nanoparticles were redispersed by the brief sonication and combined. Then, 800 μL of the resulting suspension were ultrasonicated using a probe sonicator on the ice bath (probe diameter—3 mm; amplification—60%; duration—1 min). The remaining aggregates were removed by centrifugation at 1600× *g* for 5 min. Supernatants containing nanoparticles were collected.

The size of nanoparticles was measured by dynamic light scattering (DLS). For this, nanoparticles were diluted at 1:375 in water.

### 2.2. Influence of Various Factors on the Size of ‘Artificial Peroxidase’ Nanoparticles

Synthesis of Prussian blue nanoparticles was carried out in a glass beaker under vigorous stirring (1000× *g*) with temperature control. The length of the magnet matched the diameter of the beaker (the experimental setup is shown in Appendix A). In deionized water 0.1 M solutions of FeCl_3_ and K_3_[Fe(CN)_6_] were added to 3.125 mM, then different chemical compounds were added depending on experiment: citric acid (to 2.0–5.5 mM), oxalic acid (to 1.0–4.0 mM), HCl (to pH 1 or 2), KOH (to pH 4), or KCl (to 0.1–3.0 M). In some experiments ratio of FeCl_3_ to K_3_[Fe(CN)_6_] was 5:1 (15.65 mM of FeCl_3_ + 3.125 mM of K_3_[Fe(CN)_6_]), 2:1 (6.250 mM of FeCl_3_ + 3.125 mM of K_3_[Fe(CN)_6_]), 1:1 (3.125 mM of FeCl_3_ + 3.125 mM of K_3_[Fe(CN)_6_]), 1:2 (3.125 mM of FeCl_3_ + 6.250 mM of K_3_[Fe(CN)_6_]), 1:5 (3.125 mM of FeCl_3_ + 15.65 mM of K_3_[Fe(CN)_6_]). In temperature experiments, a solution containing 3.125 mM of FeCl_3_ and K_3_[Fe(CN)_6_] was heated to +60 °C. In all other syntheses, the temperature of the reaction medium was +30 °C. Prussian blue nanoparticle deposition was initiated by 8.8 M H_2_O_2_ addition (to 22 mM). The final volume of the reaction mixture was 25 mL.

Aliquots of 500 μL were taken from the reaction mixture at different time points: after FeCl_3_ and K_3_[Fe(CN)_6_] mixing, after the addition of chemicals (if required), immediately after the H_2_O_2_ addition (0 min), and then after 10, 30, 60, 90, 120, 150 min to monitor the process of nanoparticles formation during the synthesis. The formed nanoparticles were washed with water by centrifugation at 20,000× *g* and sonicated on ice for 60 s (probe diameter—3 mm; amplification—60%; duration—1 min). Size, zeta potential, and A_700_ of nanoparticles were measured before (in situ samples) and after the washing (washed samples).

After 150 min of the synthesis, the 8 mL nanoparticles were washed by centrifugation at 20,000× *g*, redispersed in 0.8 mL of water, sonicated on ice for 60 s (probe diameter—3 mm; amplification—60%; duration—1 min), and centrifuged for 5 min at 1600× *g* or 100× *g* to remove large aggregates. The centrifugation speed depended on the synthesis method. For example, after centrifugation at 1600× *g*, sedimentation of nanoparticles larger than 200 nm was observed. Thus, suspensions of nanoparticles synthesized by the addition of citric and oxalic acids were centrifuged at 100× *g*. After that, the sizes, zeta potential, and A_700_ were measured as described below.

### 2.3. Preparation of Prussian Blue Nanoparticles with Various Sizes at 10× Scale

Prussian blue nanoparticles were prepared by reductive (5 types) and traditional (3 types) approaches. Three batches were obtained for each type of nanoparticles, except **T/dw/rt** nanoparticles, for which only two batches were prepared. The overall description of nanoparticle synthesis is given below. Details of the synthesis procedure, specific for each type are given in Table 1.

#### 2.3.1. Synthesis of Prussian Blue Nanoparticles by the Traditional Approach

Aqueous solutions of 1 mM K_4_[Fe(CN)_6_] and 1 mM FeCl_3_ were pre-heated to +55 °C and then mixed by pouring 125 mL 1 mM K_4_[Fe(CN)_6_] into 125 mL 1 mM FeCl_3_ under stirring on a magnetic stirrer. Citric acid was added to the iron salts solutions prior to their mixing if necessary. The final volume of the reaction mixture was 250 mL. The mixture was kept at +55 °C for 10 min. The solution was allowed to cool down to room temperature with stirring. NaCl was added to 1 M to nanoparticle solution, which induced their aggregation. Nanoparticles were centrifuged at 16,000× *g* until complete sedimentation; the supernatants were carefully removed, and the nanoparticle pellet was redispersed in 1 M NaCl. The washing procedure was repeated 3 times. After that, the dark blue precipitate was redispersed in 25 mL of H_2_O, ultrasonicated with 60% amplification for 30 min, and centrifuged at 1600× *g* for 5–15 min. Purification of the Prussian blue nanoparticles was carried out by dialysis in 10 kDa MWCO dialysis tubing against 2 L of deionized water. The water was changed three times. Obtained suspensions were stored at +4 °C.

#### 2.3.2. Preparation of Prussian Blue Nanoparticles by Reductive Approach (‘Artificial Peroxidase’ Nanoparticles)

In deionized water, 0.1 M solutions of FeCl_3_ and K_3_[Fe(CN)_6_] were added to 3.125 mM. Then, citric, oxalic acids, HCl, and KCl (depending on the synthesis method) were added to the required concentration. Prussian blue nanoparticle deposition was initiated by H_2_O_2_ addition (to 22 mM). The final volume of the reaction mixture was 250 mL. After 60 min the suspension was transferred into centrifuge tubes, and centrifuged at 16,000× *g* until complete sedimentation; the supernatants were carefully removed, and the nanoparticle pellet was redispersed in H_2_O by vortexing. The washing procedure was repeated 3 times. After that nanoparticles were redispersed in 25 mL of H_2_O, ultrasonicated (6 mm probe, 60% amplification), and centrifuged (Table 1). Obtained suspensions were stored at +4 °C

### 2.4. Preparation of Conjugates of Prussian Blue Nanoparticles with a Monoclonal Antibody against PSA, Protein G, and Bovine Serum Albumin

Prussian blue nanoparticles (**R2C**) were added to the gelatin A (180 bloom) aqueous solution to the final concentration of 2.41 mg/mL. The nanoparticles to gelatin mass ratio were 1:8. The resulting volume of the suspension was approximately 12 mL. The sample was vortexed, briefly sonicated (probe diameter—3 mm; amplification—60%; duration—10 s), and incubated at +37 °C for 60 min on a rotator (10 rpm). Prussian blue nanoparticles coated with gelatin A (PB/Gel A) were mixed with an equal volume of 25% glutaraldehyde (pH 7) and incubated at +37 °C for 30 min. Absorbance at 700 nm of glutaraldehyde-activated nanoparticles (further referred to as PB/Gel A-COH) was measured and used to assess nanoparticle concentration at the following synthesis stages. PB/Gel A-COH was centrifuged at 20,000× *g* until complete sedimentation. Then the precipitate was redispersed into deionized water by brief sonication (probe diameter—3 mm; amplification—60%; duration—10 s) and centrifuged at 20,000× *g* for 15 min. In total, nanoparticles were washed three times. After the final wash, nanoparticles were redispersed in a 10 mM phosphate buffer (pH 7). Then suspension of PB/Gel A-COH was divided into 3 parts. Each part was added under stirring to the solution of one of the three proteins: anti-PSA monoclonal antibodies (MAb; clone 1A6), protein G, or BSA. Mentioned proteins were preliminarily diluted in a 10 mM phosphate buffer (pH 7). PB/Gel A-COH to protein mass ratio was 1 mg to 100 μg. After that, obtained mixtures were vortexed and kept on a rotator (10 rpm) overnight at +4 °C. Glycine was added to 0.1 M to quench unreacted aldehyde groups, and the mixture was incubated at +37 °C for two more hours. Nanoparticles were washed with water as described above. After the final wash, nanoparticles were redispersed in H_2_O by sonication (probe diameter—3 mm; amplification—60%, duration—30 s). The conjugates were stored at +4 °C.

Conjugates of PB/Gel A with MAb, protein G, and BSA were labeled as PB/Gel A/MAb, PB/Gel A/protein G, and PB/Gel A/BSA, respectively.

### 2.5. Assays for PSA and Anti-Tetanus Toxoid IgG Detection

#### 2.5.1. Assay Procedure: Sandwich Immunoassay of PSA

One hundred microliters of 0.05 mg/mL mouse anti-PSA IgG (clone 3A6) in a 0.2 M carbonate buffer (pH 9.6) was added into the wells of a 96-well polystyrene plate. Plates were kept at +4 °C overnight. Plates were washed three times with 300 μL of sodium phosphate buffer with 0.1% Tween-20, pH 7 (PBT) using a microplate washer, and then 250 μL of a blocking buffer (PBT + 2% casein + 1% BSA, pH 7) was added. After 60 min of blocking, plates were washed three times. Four-fold dilutions of PSA in the blocking buffer (100 μL per well) from 1000 to 0.24 ng/mL were added, then plates were incubated for 60 min and washed three times. The PB/Gel A/MAb suspension (100 μL, 0.025 mg/mL) in the blocking buffer was added and reacted for 60 min. After washing, 100 μL of the substrate buffer (1 mL of 1 mg/mL of TMB in DMSO + 9 mL of citrate-phosphate buffer + 100 μL of 30% H_2_O_2_) was added. After 30 min, the reaction was stopped by the addition of 100 μL of 2 M sulphuric acid. The absorbance at 450 nm was measured by a microplate reader. All the assay steps except for the washing and measurement steps were performed in the thermoshaker at +37 °C (mixing speed—300 rpm).

Calibration curve was fitted to a four-parameter logistic model (1/Y^2^ weighting scheme was used). The general equation of the logistic function is:$$y=\frac{A2+\left(A1-A2\right)}{\left(1+{\left(x/x0\right)}^{p}\right)}$$

#### 2.5.2. Assay Procedure: Indirect Detection of Anti-Tetanus Toxoid IgG

One hundred microliters of 0.05 mg/mL tetanus toxoid in a 0.2 M carbonate buffer (pH 9.6) was added into the wells of a 96-well polystyrene plate. Plates were kept at +4 °C overnight. Plates were washed three times with 300 μL of sodium phosphate buffer with 0.1% Tween-20, pH 7 (PBT) using a microplate washer, and then 250 μL of a blocking buffer (PBT + 2% casein + 1% BSA, pH 7) was added. After 60 min of blocking, plates were washed three times. Four-fold dilutions of anti-tetanus antibodies in the blocking buffer (100 μL per well) from 100 mIU/mL to 0.024 mIU/mL were added, then plates were incubated for 60 min and washed three times. The PB/Gel A/Protein G suspension (100 μL, 0.025 mg/mL) in the blocking buffer was added and reacted for 60 min. After washing, 100 μL of the substrate buffer (1 mL of 1 mg/mL of TMB in DMSO + 9 mL of citrate-phosphate buffer + 100 μL of 30% H_2_O_2_) was added. After 30 min the reaction was stopped by the addition of 100 μL of 2 M sulphuric acid. The absorbance at 450 nm was measured by a microplate reader. All the assay steps except for the washing and measurement steps were performed in the thermoshaker at +37 °C (mixing speed—300 rpm). Curve fitting was performed as described in Section 2.5.1.

## 3. Results and Discussion

### 3.1. Effect of Reactants Concentration on the Size of Nanozymes

The method of obtaining ‘artificial peroxidase’ nanoparticles with improved peroxidase-like activity proposed in the original paper [9] relies on the addition of a reducing agent (H_2_O_2_) to the mixture of FeCl_3_ and K_3_[Fe(CN)_6_]. According to the mentioned paper, the size of nanoparticles can be controlled in a relatively wide range, from tens of nanometers to 300–400 nm by changing the ratio between salts and hydrogen peroxide. Unfortunately, it is not easy to predict the size of nanoparticles from a 3D graph made by the authors. Reproduction of the original method is also rather challenging, because the description of the synthesis conditions (stirring and ultrasonication regime) is not detailed enough, being however crucial for obtaining nanoparticles of the desired size. Since the mixing conditions and synthesis duration were not indicated, we synthesized nanoparticles under vigorous stirring conditions at a fixed stirring speed (1000× *g*). The concentration of iron salts was between 1.56 and 25 mM, whereas H_2_O_2_ concentration varied between 2.75 and 176 mM. Unlike the original article, synthesis was performed without the addition of HCl and KCl.

In general, larger nanoparticles were formed when the concentration of FeCl_3_/K_3_[Fe(CN)_6_] increased (Figure 1). This growth however was limited—up to 150–170 nm (expected diameter was 200–300 nm and more). Importantly, polydispersity was also relatively high (higher than the commonly accepted threshold of 0.2 [26]) at FeCl_3_/K_3_[Fe(CN)6] concentrations exceeding 6.25 mM, despite all nanoparticles being intensively sonicated and centrifuged at low speed. High polydispersity is undesirable because it negatively affects the reproducibility of nanoparticle applications and hinders control over nanoparticle processing. Notably, after prolonged post-synthesis sonication nanoparticle diameter gradually decreased until constant value (Appendix A), which indicates the importance of optimization of the post-synthesis treatment.

In total, we were not able to manipulate the size of the Prussian blue nanoparticles in the desired range by changing the ratio between iron salts and H_2_O_2_. A two-fold increase in nanoparticle size was hardly reached, moreover, the polydispersity of larger nanoparticles was too high. We admit that we did not completely follow the original protocol because HCl and KCl were not added in the course of synthesis. Nevertheless, obtained results demonstrate the non-universal nature of the size control method proposed in the original paper. Moreover, further (see Section 3.3) we show that despite both HCl and KCl can affect the size and yield of nanoparticles, they are not essential for the preparation of ‘artificial peroxidase’ with high catalytic activity.

We suggest the size of nanoparticles depends more on the stirring conditions and post-synthesis treatment (centrifugation speed, sonication duration and intensity, and so forth), rather than the concentration of iron salts. These results motivated us to study the influence of different factors including pH, temperature, ionic strength, and addition of low-molecular-weight ligands on the size and polydispersity of ‘artificial peroxidase’ nanozymes.

### 3.2. Effect of the Synthesis Conditions on the Size and Yield of Nanozymes

Alteration of synthesis parameters such as temperature [27,28], pH [29,30], presence of chelating agent [31,32,33], ratio of reactants [34], addition of organic solvent [35,36] enables manipulating the size of Prussian blue nanoparticles prepared by traditional double precursor method or hydrothermal single precursor method. We determined the influence of most of these factors on the diameter and yield of ‘artificial peroxidase’ nanozymes (Figure 2).

#### 3.2.1. Temperature

With a temperature increase from +30 to +60 °C, the mean diameter of nanoparticles grew from 103 to 152 nm (Figure 2A). Absorption at 700 nm was 15% higher for Prussian blue nanoparticles synthesized at +30 °C (Figure 2D). An increase in absorbance can be explained by both the higher yield and the smaller mean diameter [37] of obtained nanoparticles. Enlargement of nanoparticles at higher temperatures is a typical situation when reagent concentration is sufficient for both formation of new nuclei and the growth of formed particles [38].

#### 3.2.2. Chelating Agents

The addition of citric and oxalic acids decreased both starting concentration and the rate of Prussian blue formation [39] (Appendix A). This is explained by the interaction of acid molecules with ferric ions, which hinders their reaction with hexacyanoferrate ions [32]. A decrease in nuclei number and slower growth results in larger nanoparticles, however at the expense of lower yield (Figure 2E,F). An increase in citric or oxalic acid concentration resulted in a gradual increase in nanoparticle diameter (Figure 2B,C). Only negligible pH change was caused by the addition of carboxylic acids, therefore we suggest that the size increase was not due to the pH decline (see results on the influence of the pH below). Notably, a too-high concentration of carboxylic acids has the opposite effect: nanoparticles become smaller, while their yield decreases. The turning point is close to equimolar concentration. The obtained results are in line with previous reports [40,41]. Conversely, Jia et al. observed that slight growth of Prussian blue nanoparticles accompanied the addition of excessive amounts of oxalic acid [42]. Probably, the larger aging time is the reason for this result [33]. Citric acid allows for the regulation of nanoparticle size in a wider range in comparison with oxalic acid. It can be explained by the stronger complexation of iron ions by citric acid [43,44].

#### 3.2.3. Effect of pH, Ionic Strength, and Iron Salts Ratio

The pH of the reaction mixture was 2.8–2.9 and did not change during 150 min of synthesis under reference conditions (3.125 mM FeCl_3_ and K_3_[Fe(CN)_6_], 22 mM H_2_O_2_, +30 °C). Particle formation was not observed while the pH of the reaction mixture was adjusted to 4. The decrease of pH to 2 had almost no influence on the size and yield of nanoparticles (compare Figure 2A,H), whereas at pH 1 larger nanoparticles were formed (130 vs. 82 nm).

The ionic strength of the reaction mixture during synthesis under reference conditions was 37.5 mM. An increase in ionic strength led to a growth of nanoparticles from 86 nm at 0.1 M to 161 nm at 3 M (Figure 2G). The effect of ionic strength on zeta potential and polydispersity index was insignificant. A pronounced decline in nanoparticle yield (A_700_) was observed at KCl concentrations higher than 1 M (Figure 2J).

The effect of four different molar ratios of FeCl_3_ and K_3_[Fe(CN)_6_] (5:1, 2:1, 1:5, 1:2) on characteristics of Prussian blue nanoparticles was evaluated (Figure 2I,L). Excessive amounts of FeCl_3_ increased the size of nanoparticles and their polydispersity, while an excess of K_3_[Fe(CN)_6_] has no significant effect.

Relationships between the process of crystal formation and crystallization conditions such as pH or ionic strength of ions are rather complex and depend on multiple parameters, for example, the type of cations and anions [45,46]. For this reason, we are unable to present a complete explanation of observed patterns.

Based on obtained results we decided to add citric and oxalic acids into the reaction medium to obtain ‘artificial peroxidase’ nanozymes with diameters from 100 to 300 nm. The advantage of carboxylic acid addition is the possibility to obtain nanoparticles with a broad range of sizes and reasonably low polydispersity.

### 3.3. Synthesis and Characterization of Nanozymes

Five types of nanoparticles were obtained using the reductive approach (‘artificial peroxidase’) and three types of nanoparticles were synthesized by the traditional approach. Scalability and good reproducibility of nanoparticle synthesis are mandatory for their real-world application [47,48]. Therefore, for each type of nanoparticles, 3 individual batches were synthesized (only 2 were prepared for **T/dw/rt**). Starting reaction volume was increased tenfold in comparison with previous experiments: from 25 to 250 mL.

Based on previous results, four types of conditions were chosen to synthesize nanoparticles with diameters from 100 to 300 nm by the reductive approach. Citric and oxalic acid were utilized to control the diameter of nanoparticles. One more type of nanoparticles was prepared by the reductive approach using conditions (0.1 M KCl and 0.1 M HCl) proposed in the original paper by Komkova et al. [9]. Additionally, three types of Prussian blue nanoparticles were synthesized using the traditional approach, their sizes varied from 60 to 130 nm. Characteristics and naming of all types of nanoparticles can be found in Table 1 and Appendix A.

#### 3.3.1. Size and Morphology

Preparation of large nanoparticles by the traditional method is a rather challenging task. The largest size of nanoparticles we were able to reach (120–130 nm) was achieved by the very slow addition of K_4_[Fe(CN)_6_] to the FeCl_3_ solution (Figure 3A). It was important to prepare such large nanoparticles because we were interested in comparing nanoparticles of similar size prepared by two different approaches. Unfortunately, we failed to synthesize ‘artificial peroxidase’ nanoparticles smaller than 90 nm, therefore batches **R**, **T**, **R2C**, and **T/dw/rt** were the only ones with similar sizes.

TEM analysis showed that “artificial peroxidase” nanoparticles have an irregular shape and are rather disordered assemblies of smaller nanoparticles (Appendix A). The irregular shape of nanoparticles made it impossible to achieve an accurate measurement of their diameter by TEM. In sample **RKH** along with large nanoparticles, small cubic nanoparticles were also observed (Appendix A). These factors together explain the relatively high polydispersity (no samples with PdI lower than 0.1) of ‘artificial peroxidase’ nanoparticles (size distribution plots are given in Appendix A).

In general, the size of ‘artificial peroxidase’ nanoparticles was close to the target size (nanoparticles prepared in identical conditions in 25 mL reaction volume). The difference between target size and observed size was in the range from −9% to +11%. Therefore, the addition of carboxylic acids allows for properly controlling the size of nanoparticles prepared by the reductive approach. The reproducibility of nanoparticle preparation was examined by estimating the batch-to-batch coefficient of variation for mean hydrodynamic diameter. The coefficient of variation was between 0.3% and 2.6% for “artificial peroxidase” nanoparticles, indicating good batch-to-batch reproducibility. For comparison, between-batch coefficients of variation reported for other nanoparticles were 2–5% (gold nanoparticles) [49], 10–42% (dextran-coated iron oxide nanoparticles) [50], less than 5% (poly(D,l-lactide-co-glycolide) nanoparticles) [51].

#### 3.3.2. Yield

The highest yield (Figure 3C) was observed when no additives were present in the reaction medium (R)—approximately 200 mg of nanoparticles were isolated, which is in agreement with data on absorbance at 700 nm obtained in small-scale experiments (Figure 2A,D). Both citric and oxalic acid decreases the concentration of nanoparticles. In general, the larger the diameter of nanoparticles, the lower the yield. The presence of KCl and HCl (as in the original paper) resulted in a two-fold lower yield in comparison with the **R2C** sample, despite sizes being almost the same.

#### 3.3.3. Elemental Analysis

The concentration of iron and potassium was determined for each batch by the ICP-MS. Iron content was used to normalize peroxidatic activity of Prussian blue nanoparticles synthesized by various methods (see below in this section). Depending on the potassium concentration, two types of Prussian blue can be distinguished: soluble (contains potassium ions) and insoluble (does not contain potassium ions) [52]. Insoluble Prussian blue contains Fe^II^(CN)_6_^4−^ vacancies, filled with water molecules that are partially coordinated to Fe^III^ ions. Soluble Prussian blue contains potassium ions in interstitial sites of Prussian blue cubic structure. In this case, the number of vacancies depends on potassium content [52]. All Prussian blue nanoparticles prepared by the traditional method consisted of insoluble Prussian blue, whereas ‘artificial peroxidase’ nanozymes contained sufficient amounts of potassium. Fe:K mass ratio was from 17.6 (**RKH**) to 66.8 (**R2C**). **RKH** nanoparticles were more saturated with potassium because they were synthesized in the presence of 0.1 M KCl. Opposite to the previous report [53], we did not observe a clear relationship between potassium content and catalytic activity of nanozymes (Figure 3D,F,G). Even if some dependence exists, in this study, it was masked by other factors, first, the size difference between nanoparticles prepared in various conditions.

#### 3.3.4. The Catalytic Activity of Prussian Blue Nanozymes

The most important feature of ‘artificial peroxidase’ nanozymes is improved peroxidatic activity in comparison with Prussian blue nanoparticles synthesized by the traditional method. Komkova et al. [9] calculated the catalytic activity of ‘artificial peroxidase’ considering a single nanoparticle as a catalytic unit. The issue of measurement of the nanozymes’ catalytic activity has been recently discussed by Zandieh and Liu [54]. They recommended calculating the number of catalytically active sites, e.g., surface iron atoms in non-porous iron oxide nanoparticles, to avoid over- or underestimation of catalytic activity. However, an accurate determination of the number of Prussian blue nanoparticles or surface iron atoms is difficult, because synthesized nanoparticles despite having a relatively narrow size distribution are not monodisperse. Besides, according to TEM analysis, their shape is irregular. Interaction with substrates is one more factor that complicates the quantification of catalytically active metal sites in Prussian blue nanozymes. Catalytic mechanism of ‘artificial peroxidase’ includes reduction of nanozyme to prussian white by TMB or another substrate, which is followed by oxidation of prussian white with H_2_O_2_ [55]. Substrate molecules are able to penetrate the nanoparticle body being able to evolve inner metal atoms into a catalytic process [9], however, estimation of catalytically active metal sites is hardly possible as depends on the particle diameter. Therefore, we chose a more straightforward approach and measured catalytic activity normalized to the iron concentration (units per milligram of iron, U/mgFe).

Catalytic activity was expressed in terms of specific activity units, i.e., the number of moles of a substrate that are converted to product by one milligram of catalyst in one min. This measure was used because the determination of specific activity is conducted in the conditions of substrate excess that resembles the detection step in colorimetric analyses. Indeed, nanozymes with increased specific activity provided a lower limit of detection in colorimetric immunoassay [56]. The whole procedure of specific activity determination was adapted from Jiang et al. [57], however, buffer composition was preliminary optimized. The decreased ionic strength of the reaction buffer prevented nanoparticle aggregation. The concentration of hydrogen peroxide was also lowered to 100 mM to achieve better reproducibility. The specific activity of horseradish peroxidase was determined in the same conditions. Undoubtedly, experimental conditions were suboptimal for horseradish peroxidase, because in conventional ELISA horseradish peroxidase reacts in the presence of 1–5 mM of H_2_O_2_. Higher H_2_O_2_ can decrease the specific activity of enzymes due to heme destruction. Optimal conditions for nanozymes (buffer composition, ionic strength, pH [58]) were also not optimized, because our primary goal was to compare nanozymes synthesized in different conditions rather than maximize their performance. The issue of comparison between enzymes and nanozymes is complex and outside the scope of this work, however, even in non-optimal conditions horseradish peroxidase outperformed nanozymes (Figure 3F).

Prussian blue nanoparticles obtained by the reductive approach (‘artificial peroxidase’) exhibited an equal or higher peroxidase-like activity in comparison with nanoparticles synthesized by the traditional method having similar size (**R** vs. **T**: 2.46 ± 0.08 U/mgFe vs. 2.06 ± 0.019 U/mgFe, *p* = 0.079, and **R2O** vs. **T/dw/rt**: 3.13 ± 0.07 U/mgFe vs. 1.85 ± 0.06 U/mgFe, *p* < 0.001; Figure 3F and Appendix A). Moreover, the catalytic activity of **R2C** and **R2O** exceeds that of **T** and **T/dw/rt**, which have much smaller hydrodynamic diameters (Appendix A). **RKH** is equally active to **T** and **T/dw/rt**, which are almost two-fold lower. As a rule, smaller nanoparticles have higher catalytic activity due to larger specific surface area [59], however, there was no clear dependence between size, surface area, and peroxidatic activity of nanozymes. (Figure 3A,E,F). Unexpectedly, nanoparticles with the highest catalytic activity (**R2O**) had the lowest specific surface area. The largest nanoparticles (**R4.5C**) had the lowest catalytic activity, however, the smallest ones (**R**) were not the most active. Nanoparticles, synthesized in the presence of 2 mM oxalic acid (**R2O**) showed the best catalytic performance.

In general, nanoparticles prepared with the addition of carboxylic acid had better activity in comparison with nanoparticles synthesized according to the original method (**RKH**). Probably, carboxylic acids somehow enhance the catalytic activity of Prussian blue nanoparticles. Recently, Feng et al. prepared Prussian blue nanoparticles in the presence of citric acid with various sizes and crystallinity by regulating the regime of reagents addition [33]. Authors demonstrated that nanoparticles that are more amorphous possessed better peroxidase- and catalase-like activity, which is also true for other nanomaterials [60]. However, these nanoparticles were the smallest ones and it was not clear whether their activity stems from a small size or structural defects. In our study, highly active nanoparticles prepared by carboxylic acids-assisted processes (**R2C** and **R2O**) were larger or similar to nanoparticles synthesized in the absence of acids. Therefore, we suggested that they have a more amorphous structure in comparison with other nanoparticles because in previous studies addition of citrate in the course of synthesis led to the appearance of internal and external defects in gold nanoparticles [61] and Prussian blue analogs [28,62]. On the other hand, citric acid facilitated the preparation of Prussian blue nanoparticles with perfect crystallinity [40]. Evidently, the morphology of Prussian blue nanoparticles depends on both concentrations of chelating agent and aging time [33,63]. Synthesis in the presence of hydrogen peroxide also decreased the crystallinity of Prussian blue analogs [64]. Indeed, selected area electron diffraction (SAED) patterns obtained for synthesized nanoparticles exhibited mostly the amorphous structure of ‘artificial peroxidase’ nanozymes (Appendix A). However, no significant differences in crystallinity were revealed for different types of ‘artificial peroxidase’. Surface charge also affects the catalytic activity of nanozymes by regulating interactions with the substrate [65]. The zeta potential of all Prussian blue nanoparticles was less than −30 mV (Figure 3B) facilitating adsorption of positively charged TMB. We did not observe any relationship between zeta potential and the specific activity of nanozymes.

Taken together, our results indicate that ‘artificial peroxidase’ nanozymes have an amorphous structure, beneficial for their catalytic activity. The reasons underlying the difference in catalytic activity between ‘artificial peroxidase’ nanozymes synthesized in various conditions and nanoparticles prepared by the traditional approach are not clear and require more detailed study.

#### 3.3.5. Storage Stability

In terms of the preparation of diagnostic reagents, it is convenient to store nanoparticle preparations that are readily available for conjugation. From our point of view, concentrated aqueous suspensions of Prussian blue nanoparticles are the most suitable form for long-term storage. Komkova et al. [9] suggested storage of ‘artificial peroxidase’ in a dry state or in 0.1 M KCl + 0.1 M HCl solution in order to prevent hydrolysis, however, the latter approach requires preliminary removal of acid and salt. This can be challenging because smaller nanoparticles (less than 100 nm) require very high centrifugation speed, whereas other desalting methods such as dialysis or gel filtration are time-consuming. Dry nanoparticles can be easily diluted in the medium of interest, however, drying can affect their size, and therefore additional ultrasound treatment may be necessary. In this work, Prussian blue nanoparticles were stored in deionized water for 5 months. It has been shown that the size of nanoparticles did not change during three months for all types of syntheses (Figure 3H, Appendix A). After three months of storage, aggregation was observed in some batches. Aggregates were removed by nanoparticle ultrasound treatment and low-speed centrifugation. Nevertheless, these additional manipulations are time-consuming, therefore the storage period of nanozyme suspensions should not exceed 3 months. One should note that we did not examine the stability of dried or freeze-dried nanoparticles.

### 3.4. Conjugation of ‘Artificial Peroxidase’ Nanoparticles with Recognition Molecules and Their Application in ELISA-like Immunoassay

We functionalized ‘artificial peroxidase’ nanoparticles with *Streptococcal* protein G, monoclonal antibodies against the prostate-specific antigen, and BSA by both covalent attachment and adsorption.

For covalent attachment ‘artificial peroxidase’ nanoparticles were coated with gelatin A, then the gelatin layer was cross-linked with glutaraldehyde. Primary amines of protein G were reacted with free aldehyde groups (Figure 4). The gelatin coating was chosen for several reasons. First, gelatin provides good colloidal stability over a wide range of pH values including acidic ones [66], which is significant, because hydrolysis of Prussian blue occurs at neutral and alkaline pH (see below in this section) [67]. Second, gelatin is a commonly available non-toxic polymer containing multiple functional groups. Gelatin A has a pI of 7–9, being positively charged at neutral and acidic pH and, therefore, readily adsorbed to negatively charged nanoparticles (Appendix A). Gelatin cross-linking and binding of affine molecules was carried out at pH 7 because glutaraldehyde reacts with primary amines at neutral and alkaline pH [68]. One should note that a sufficient amount (approximately 40%) of Prussian blue nanoparticles were lost due to hydrolysis after overnight incubation at pH 7 (measured by A_700_ difference).

Successful functionalization was confirmed by measuring colloidal stability, zeta potential, and functional activity of nanoparticles. Alteration of zeta potential reflects the change of nanoparticle surface structure (Figure 5A). Uncoated nanoparticles have a strong negative charge (−36.3 mV), whereas after coating with gelatin zeta potential value becomes more positive: −0.1 mV. Cross-linking results in the decrease of zeta potential (−5.6 mV) which is explained by lowering the number of surface primary amines. Attachment of affine molecules and blocking with glycine changed zeta potential to more positive values. In contrast to non-coated Prussian blue nanoparticles, gelatin-modified ones were stable in McIlvaine buffer with pH from 3 to 7 after 60 min of incubation (Figure 5B). Colloidal stability of nanoparticles did not depend on zeta potential value, being therefore provided by the steric repulsion or other forces (Figure 5C). Protein coating decreased the catalytic activity of nanoparticles more than three-fold (Appendix A). Evidently, gelatin molecules hinder the interaction of the substrate with nanoparticle catalytic sites. The morphology of nanoparticles was studied by TEM (Figure 5E,F). Nanoparticles have irregular shapes, and their sizes were substantially different, which is not in line with DLS data, which reported low polydispersity of nanoparticles. The possible reason is an underestimation of smaller nanoparticles by DLS, which is a well-known feature of this method [69].

As was mentioned before, hydrolysis of Prussian blue occurs at neutral and acidic pH. A decline of A_700_ was observed when PB/Gel A/Protein were stored for 24 h in various buffers solutions with pH from 2 to 8. (Appendix A). The decrease of A_700_ was more prominent at pH 6–8. In some samples, this can be explained by nanoparticle aggregation (Appendix A), but in most of them, the size of nanoparticles changed insignificantly. Conversely, in deionized water, the decline of A_700_ did not exceed 15% in 7 months (Appendix A). Therefore, functionalized nanozymes should be stored in deionized water for several months and diluted in assay buffer (for example, TRIS, HEPES, or phosphate) prior to assay. In this case, the effect of hydrolysis will be negligible. Additional comments on the storage stability of nanozymes can be found in the Appendix A.

Apart from covalent conjugation, ‘artificial peroxidase’ nanoparticles were non-covalently functionalized with protein G and BSA (negative control). For this, nanoparticles were incubated with affine molecules, followed by blocking of unoccupied sites with an excessive amount of gelatin A. Antibody-to-nanoparticle ratio, buffer composition, and washing procedures were similar to that of covalent functionalization. During functionalization nanoparticles aggregated (Appendix A), and they generated high non-specific signal (Appendix A), being therefore inappropriate for immunoassay.

The practical applicability of PB/Gel A/Mab and PB/Gel A/Protein G as diagnostic reagents for colorimetric immunoassay was proven by quantitative detection of two model analytes: prostate-specific antigen (PSA) and anti-tetanus IgG. The limit of detection (LOD) was determined as the concentration of analyte, which corresponds to the mean absorbance value of zero calibrators plus three standard deviations. For the PSA assay, LOD was 0.189 ng/mL, for tetanus toxoid—0.035 mIU/mL. In experiments where nanoparticles conjugated with monoclonal antibodies and protein G were replaced with control nanoparticles (conjugated with BSA), only a slight increase of absorbance at the highest analyte levels was observed (see Figure 6(B-2,C-2), and blue curves in Figure 6D,E). This result demonstrates that the ability of nanoconjugates to interact with analytes is due to the presence of affine molecules on their surface. Anyway, even though we did not perform complete systematic optimization of immunoassay procedures, both of them allow the detection of analytes at concentrations, which are relevant for clinical application. For PSA it is a concentration range from 0.1–0.2 to more than 20 ng/mL [70], for tetanus IgG tests should be able to detect antibodies from the lowest protective titers (10–100 mIU/mL) to the highest titers in immunized individuals (several U/mL) [71]. One can see that sensitivity of developed immunoassay is enough to detect PSA in undiluted blood serum/plasma, whereas detection of anti-tetanus IgG is possible even in highly diluted specimens. Undoubtedly, the real clinical value of the ‘artificial peroxidase’-based assays can be assessed only after their strict optimization and validation. Besides, despite sufficient sensitivity, presented assays do not outperform commercial ELISAs, which allow detection of the same analytes at lower concentrations (for example, https://www.abcam.com/human-psa-elisa-kit-ab264615.html (accessed on 10 February 2022)).

## 4. Conclusions

Nanozymes ‘artificial peroxidase’ are a promising alternative to natural enzymes, due to high catalytic activity, facile synthesis procedure from cheap reagents, and satisfactory storage stability. In this work, we developed a robust and reproducible method for the preparation of ‘artificial peroxidase’ nanozymes with tunable size. The method relies on the addition of citric and oxalic acids which change the size of nanoparticles by decreasing the reaction rate. Besides, we demonstrated that tuning various synthesis parameters such as temperature, pH, and salt concentration could also be used to manipulate the properties of ‘artificial peroxidase’ nanozymes. In total, our results confirm that Prussian blue nanoparticles prepared by the reductive approach have higher peroxidatic activity compared to those prepared by the traditional method. Moreover, synthesis in the presence of carboxylic acids enables the preparation of ‘artificial peroxidase’ with enhanced peroxidatic activity in comparison with nanozymes synthesized by the original method [9].

Although we synthesized nanozymes in tens- to hundreds-of-milligram-scale, further optimization and scale-up are necessary. Nowadays, significant attention is paid to the large-scale preparation of nanoparticles with the aid of micro-and nanofluidics [48]. Indeed, flow synthesis of ‘artificial peroxidase’ has been recently developed [72]. Panariello and colleagues demonstrated that in some circumstances conditions of nanoparticle synthesis might be directly translated from batch to flow reactors [49], therefore we believe that our data on the synthesis process can be employed during the preparation of Prussian blue nanozymes in flow systems.

Coating of Prussian blue nanoparticles with a gelatin shell endows them with excellent colloidal and shelf-life stability. Our results demonstrate that optimization of Prussian blue nanoparticle coating is important not only in terms of their colloidal stability, non-specific interactions in immunoassay, and attachment of affine molecules but also in terms of their stability to hydrolysis in neutral and alkaline conditions. We suggest that an assessment of hydrolysis intensity needs to be performed for all Prussian blue nanoparticles encountering physiological media (buffers, blood serum or plasma, cultural fluids, and so on) or alkaline solutions.

Diagnostic reagents for colorimetric immunoassay were prepared from gelatin-coated ‘artificial peroxidase’ nanozymes. They allowed successful detection of model analytes demonstrating suitability for real-world applications. Nevertheless, high efficiency in immunoassay and storage stability alone are not enough to consider ‘artificial peroxidase’ nanozymes as competitors of enzymes in the field of colorimetric assays [6]. Nanozymes should propose significant advantages compared to enzymes, because commercial enzyme-based assays possess excellent analytical characteristics, and are technologically mature and well optimized. Issues of high-throughput synthesis and hydrolysis stability of ‘artificial peroxidase’ nanozymes despite being partially addressed [72,73], however still to be completely resolved. Further improvement of catalytic activity via defect engineering [74] or active site microenvironment [75] is also necessary.

## Figures and Tables

**Figure 1 nanomaterials-12-01630-f001:**
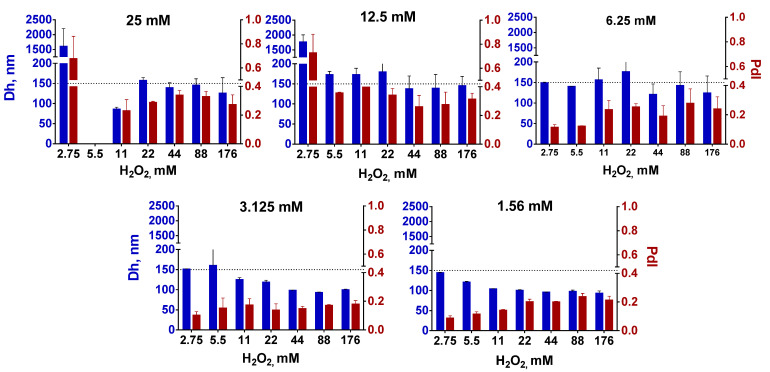
Size and polydispersity of Prussian blue nanoparticles prepared at different concentrations of FeCl_3_/K_3_[Fe(CN)_6_] (shown above the histograms) and H_2_O_2_. Dh—hydrodynamic diameter (blue bars), PdI—polydispersity index (claret red bars). The dotted line at 150 nm is for easier visual comparison of samples. The absence of a bar means an aggregation of nanoparticles. *n* = 3, mean ± SD.

**Figure 2 nanomaterials-12-01630-f002:**
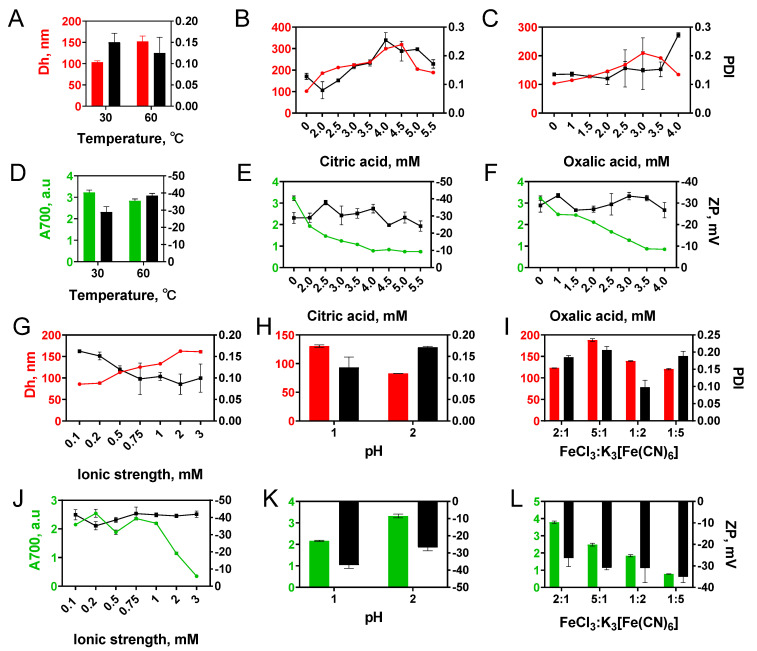
Influence of different factors: (**A**,**D**) temperature, (**B**,**E**) citric acid concentration, (**C**,**F**) oxalic acid concentration, (**G**,**J**) ionic strength, (**H**,**K**) pH, (**I**,**L**) molar ratios of FeCl_3_ and K_3_[Fe(CN)_6_] on the size (Dh), polydispersity (PDI) of nanoparticles, yield (A700), and zeta potential (ZP) of ‘artificial peroxidase’ nanozymes. *n* = 3, mean ± SD.

**Figure 3 nanomaterials-12-01630-f003:**
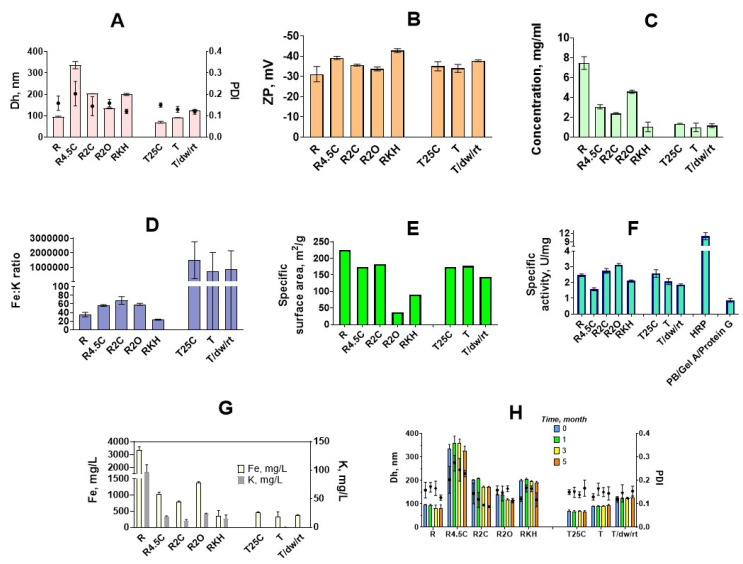
Characterization of prussian blue nanoparticles synthesized at 10x scale. (**A**) mean hydrodynamic diameter (Dh; bars) and polydispersity index (PDI); (**B**) zeta potential (ZP) of obtained nanoparticles; (**C**) concentrations (dry mass per unit volume); (**D**) Fe:K mass ratio; (**E**) specific surface area; (**F**) peroxidase-like activity (HRP—horseradish peroxidase); (**G**) concentration of Fe (white bars) and K (gray bars); (**H**) size (Dh; bars) and polydispersity (PDI) of nanoparticles after 1, 3, and 5 months of storage at +4 °C. *n* = 3, mean of three individual batches ± SD (for E, *n* = 1).

**Figure 4 nanomaterials-12-01630-f004:**
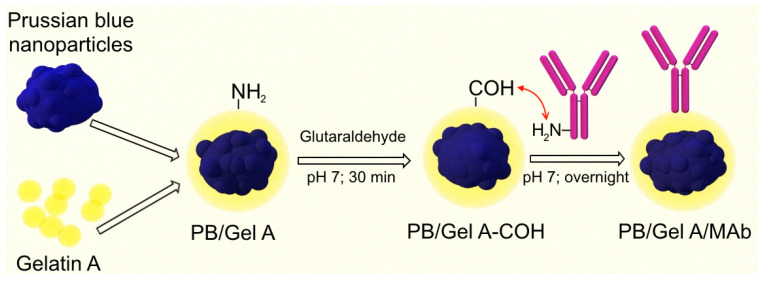
Conjugation of gelatin-coated Prussian blue nanoparticles with monoclonal antibodies. Gelatin amine groups were cross-linked with glutaraldehyde. Then, amine groups of antibodies were reacted with the surface carbonyl groups.

**Figure 5 nanomaterials-12-01630-f005:**
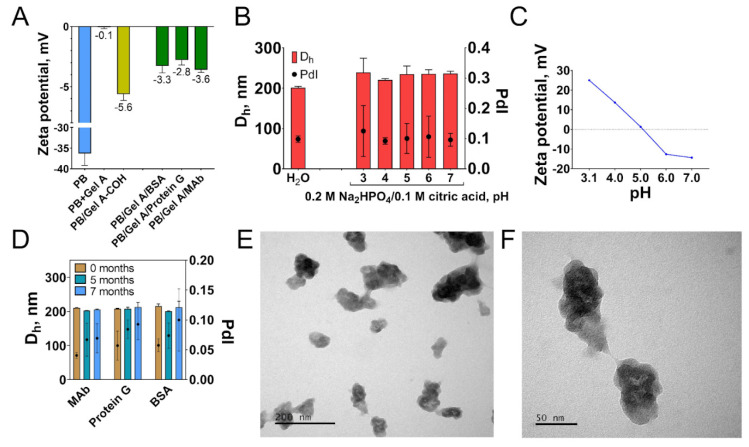
(**A**) Change of nanoparticle zeta potential in the course of conjugation. (**B**) Colloidal stability of PB/Gel A/Protein G in water and McIlvaine buffer with pH 3–8 (60 min incubation). (**C**) Zeta potential of PB/Gel A/Protein G in 5 mM Na_2_HPO_4_/citric acid buffer, pH 3.1–7. (**D**) Long-term colloidal stability of PB/Gel A/MAb, PB/Gel A/Protein G, and PB/Gel A/BSA in water. (**E**,**F**) TEM images of PB/Gel A/MAb. Scale bars: e—200 nm, f—50 nm. *n* = 3, mean ± SD.

**Figure 6 nanomaterials-12-01630-f006:**
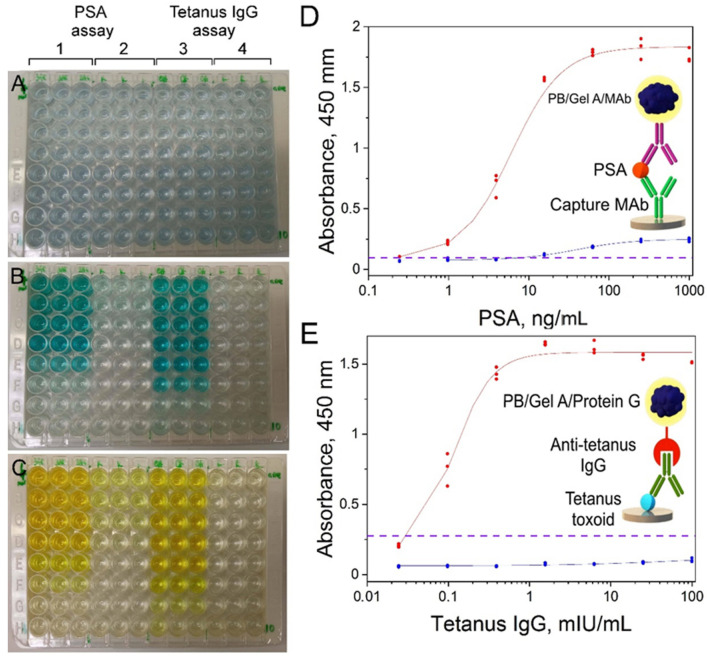
Colorimetric assay of PSA and anti-tetanus IgG. (**A**–**C**) polystyrene plate at different stages of the assay; (**D**,**E**) corresponding dose-response curves. (**A**) plate filled with conjugates of Prussian blue nanoparticles, (**B**) plate after 30 min of reaction with substrate, (**C**) plate after the addition of 2 M H_2_SO_4_. Concentration of PSA and anti-tetanus IgG decreased from top to bottom row, 1—wells filled with PB/Gel A/MAb, 3—wells filled with PB/Gel A/Protein G, 2 and 4—wells filled with PB/Gel A/BSA (negative control). Red lines—sigmoidal fit of PSA and anti-tetanus IgG calibration curves. Blue lines—dose-response curves obtained using PB/Gel A/BSA. Purple dashed line at absorbance value equal to mean absorbance of the zero calibrator + 3 × standard deviation. *n* = 3.

**Table 1 nanomaterials-12-01630-t001:** Details of Prussian blue nanoparticle synthesis (scale-up experiment).

Designation ^a^	Synthesis Conditions
**R**	**FeCl_3_ и K_3_[Fe(CN)_6_]**—3.125 мM; **H_2_O_2_**—22 мM; +30 °C, stirring—1000× *g*, sonication—40 min, final centrifugation at 1600× *g* for 15 min
**R4.5C**	**FeCl_3_ и K_3_[Fe(CN)_6_]**—3.125 мM; **Citric acid**—4.5 mM; **H_2_O_2_**—22 мM; +30 °C, stirring—1000× *g*, sonication—60 min, final centrifugation at 100× *g* for 5 min
**R2C**	**FeCl_3_ и K_3_[Fe(CN)_6_]**—3.125 мM; **Citric acid**—2mM; **H_2_O_2_**—22 мM; +30 °C, stirring—1000× *g*, sonication—60 min, final centrifugation at 100× *g* for 15 min
**R2O**	**FeCl_3_ и K_3_[Fe(CN)_6_]**—3.125 мM; **Oxalic acid**—2 mM; **H_2_O_2_**—22 мM; +30 °C, stirring—1000× *g*, sonication—40 min, final centrifugation at 100× *g* for 15 min
**RKH**	**FeCl_3_ и K_3_[Fe(CN)_6_]**—3.125 мM; HCl—0.1 M; KCl—0.1 M; **H_2_O_2_**—22 мM; +30 °C, stirring—1000× *g*, sonication—60 min, final centrifugation at 1600× *g* for 10 min
**T25C**	**FeCl_3_ и K_4_[Fe(CN)_6_]**—1 мM; **Citric acid**—25 mM; +55 °C, stirring—1000× *g*, sonication—30 min, final centrifugation at 1600× *g* for 10 min
**T**	**FeCl_3_ и K_4_[Fe(CN)_6_]**—1 мM; +55 °C, stirring—1000× *g*, sonication—30 min, final centrifugation at 1600× *g* for 10 min
**T/dw/rt**	**FeCl_3_ и K_4_[Fe(CN)_6_]**—1 мM; the solution of K_4_[Fe(CN)_6_] was added to the FeCl_3_ solution dropwise with a rate of 10 mL/h using a peristaltic pump; room temperature, stirring—7000× *g*, sonication—30 min, final centrifugation at 1600× *g* for 10 min

^a^ R—synthesis of Prussian blue nanoparticles based on reduction of a mixture of ferricyanide and ferric ions by hydrogen peroxide, C—citric acid, O—oxalic acid, T—traditional synthesis of Prussian blue nanoparticles, dw—dropwise, rt—room temperature.

## Data Availability

The datasets used and/or analyzed during the current study are available from the corresponding author on reasonable request.

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
