# Peer review of "Prussian Blue Nanozymes with Enhanced Catalytic Activity: Size Tuning and Application in ELISA-like Immunoassay"

_nanomaterials, 2022, doi:10.3390/nano12101630_

Round 1

Reviewer 1 Report

To date several methods to synthesize Prussian Blue Nanoparticles are available. This manuscript entitled “Prussian blue nanozymes with enhances catalytic activity: size tuning and application in ELISA-like immunoassay” by Khramtsov. et al. described a reproducible and scalable method for the preparation of nanozymes with catalytic properties of 'artificial peroxidase' based on previous reports described in the literature. The authors study the influences of the parameters to determine the optimal condition in the size and yield of the nanozymes in the synthesis. In addition, the activity of coated ‘nanozymes was applied as efficient peroxidase mimics for colorimetric immunoassay.

This reviewer thinks that this manuscript should be publishable to a filed-specific journal instead of in Journal of Nanomaterials

I think the end of the introduction is not with a good style for scientific publication. The methodology is also not clear and confusing.

I think that the authors need to show the synthetic methodology of the each nanomaterial synthesized with good characterization, including in all cases TEM. The only TEM images that showed to me are particle agglomeration or or non-homogeneous nanoparticles.The use of DLS without showing each spectrum with the different sized particle populations - if any - in the measurement is not enough to optimise nanoparticle synthesis methods. Also, for the application in the field of colorimetric assays, the advantages of the catalytic nanozymes developed in this study against their competitors of nanozymes of different metals-based nanoparticles should be specified with more details. 

Author Response

Reviewer #1

We express our great gratitude to the Reviewer for comments and thoughtful suggestions. Based on these comments and suggestions, we have made modifications to the original manuscript. In general, we made the manuscript more concise. Excessive information was removed or transferred to Supplementary Materials.

Key changes we made are summarized below:

  1. Last part of the Introduction was rewritten.
  2. Methods of nanoparticles characterization were moved to Supplementary Materials.
  3. Methodological details were removed from the Results and Discussion, except for Section 3.3 (first two paragraphs), where it is necessary for better understanding of the experiment design.
  4. TEM and SAED results for all samples were added (they can be found in the Supplementary Materials, figures S5-7). Data on specific surface area of nanoparticles was added (Fig 3E). Effect of protein coating on specific activity of nanozymes was studied (Section 3.4. and Figure S10).
  5. Curve fitting was performed using 4-parameter logistic model, according to the recommendations of the Referee. Details of curve fitting were added to the section 2.5.

All changes are highlighted with yellow.

We believe that the manuscript has been greatly improved and hope it has reached high journal's standard.

  1. I think the end of the introduction is not with a good style for scientific publication.

Response:

The last part of the Introduction was rewritten.

  1. The methodology is also not clear and confusing.

Response:

Experimental section was changed. Excessive tables and figures as well as Section on nanoparticles characterization were moved to the Supporting Information.

  1. I think that the authors need to show the synthetic methodology of the each nanomaterial synthesized with good characterization, including in all cases TEM

Response:

For each type of nanoparticles synthetic methodology is thoroughly described including reactants concentration, conditions (temperature, mixing speed, time of incubation), purification, and post-synthesis treatment (sonication regime, removal of aggregates by centrifugation). Methodology can be found in the Section 2.3 and Table 1. Nanoparticles were characterized by several methods including ICP-MS, TEM, SAED, DLS, M3-PALS (zeta-potential), nitrogen adsorption (specific surface area). Catalytic activity and storage stability of all nanoparticles were assessed. TEM images of all synthesized nanoparticles can be found in Supporting Information (Figures S5 and S6).

  1. The only TEM images that showed to me are particle agglomeration or or non-homogeneous nanoparticles.

Response:

We agree, that nanoparticles have irregular shape and large aggregates can be seen in TEM images. Probably, this can be explained by both presence of large aggregates in nanoparticle suspensions and aggregation of nanoparticles upon drying in the course of TEM samples preparation. Taking into account that most of nanoparticles are relatively monodisperse (polydispersity index is less than 0.2), we suppose that presence of large aggregates in TEM images is rather result of drying. Dynamic light scattering is sensitive to the presence of large nanoparticles, therefore one can expect that large aggregates in suspension will lead to high polydispersity indices, but this is not the case. Nevertheless, nanozyme suspensions contain some percentage of aggregates and nanoparticle shape is not uniform, especially in comparison with best examples, such as quantum dots or gold nanoparticles, which can be extremely uniform. Besides, in paper we emphasized that we had some issues with between-replicates reproducibility of DLS measurements, cased by the presence of such aggregates (in the revised version of manuscript these data were moved to Supporting Information, Table S4). We consider preparation of homogeneous “artificial peroxidase” nanoparticles as an issue which will be solved in future studies, maybe by polymer-assisted synthesis.

  1. The use of DLS without showing each spectrum with the different sized particle populations - if any - in the measurement is not enough to optimise nanoparticle synthesis methods.

Response:

Please, find size distribution plots (obtained by DLS) in Supporting Information, figure S7. Most of nanoparticles had only one large peak. In some cases small peak in the area of micrometer-sized nanoparticles was detected, however as a rule these large-sized aggregates were not detected in all the technical replicates (please, see Table 4 for details).

  1. Also, for the application in the field of colorimetric assays, the advantages of the catalytic nanozymes developed in this study against their competitors of nanozymes of different metals-based nanoparticles should be specified with more details.

Response:

Advantages of ‘artificial peroxidase’ nanozymes are summarized in the beginning of the Conclusion. They include high catalytic activity, non-expensive precursors, straightforward synthesis, and good stability (both colloidal and long-term). Direct comparison of prussian blue nanozymes with the most active nanozymes, such as single-atom nanozymes (Anal. Chem. 2021, 93, 3, 1221–1231), Ni-Pt nanoparticles (J. Am. Chem. Soc. 2021, 143, 7, 2660–2664), or any other is a quite complex task, because they have different structure (shape, composition, specific surface area etc.). Even simple comparison of catalytic activity is not as easy as can be expected (challenges are discussed in ACS Nano 2021, 15, 10, 15645–15655 and in Langmuir 2022, 38, 12, 3617–3622). All these materials are synthesized by the completely different methods, therefore comparison of technological aspects is also hardly possible. We suppose that comparison of various catalytic nanomaterials requires separate set of experiments. Without experiments, such comparison has a limited value.

Reviewer 2 Report

In this work, Khramtsov et al reported size-tunable Prussian blue (PBs) as nanozymes, and they further used modified PBs to design immunoassays for PSA detection. Compared to previous methods, current method is more reliable and scalable to prepare PB nanoparticles with size from 100 to 300 nm. The authors found that several environmental parameters may affect the synthesis of PB nanozymes. Overall, this work provides a detailed study on the reproducible production of an important nanozyme and its application in bioassays. The reviewer would like recommend the work for publication after addressing some minor issues.

  1. The authors might need to write the paper more concisely. Current manuscript is wordy with too much unnecessary details. Some figures or tables can be moved to the supporting information. For example, details regarding the synthesis procedures have already been described in the methods section. The authors should refer to the data quickly in each section.
  2. Figure 4 contains too much information that one could barely understand the way of organization.
  3. The authors might need to shorten the subheadings (e.g. 3.1, 3.2).
  4. When designing immunoassays, the authors coated that NPs with a gelatin layer. Nanozymes largely rely on their surface active sites to perform catalytic reactions. Does such modifications affect the nanozymatic activity of PBs?

Author Response

Reviewer #2

We express our great gratitude to the Reviewer for comments and thoughtful suggestions. Based on these comments and suggestions, we have made modifications to the original manuscript. In general, we made the manuscript more concise. Excessive information was removed or transferred to Supplementary Materials.

Key changes we made are summarized below:

  1. Last part of the Introduction was rewritten.
  2. Methods of nanoparticles characterization were moved to Supplementary Materials.
  3. Methodological details were removed from the Results and Discussion, except for Section 3.3 (first two paragraphs), where it is necessary for better understanding of the experiment design.
  4. TEM and SAED results for all samples were added (they can be found in the Supplementary Materials, figures S5-7). Data on specific surface area of nanoparticles was added (Fig 3E). Effect of protein coating on specific activity of nanozymes was studied (Section 3.4. and Figure S10).
  5. Curve fitting was performed using 4-parameter logistic model, according to the recommendations of the Referee. Details of curve fitting were added to the section 2.5.

All changes are highlighted with yellow.

We believe that the manuscript has been greatly improved and hope it has reached high journal's standard.

  1. The authors might need to write the paper more concisely. Current manuscript is wordy with too much unnecessary details. Some figures or tables can be moved to the supporting information. For example, details regarding the synthesis procedures have already been described in the methods section. The authors should refer to the data quickly in each section.

Response:

Excessive information such as methodological aspects and discussion of insignificant or unrelated results were removed or partially transferred to the Supporting Information. Methods of nanoparticles characterization and some other methods (including some figures and tables) were also moved to the Supporting Information. Sections on optimization of synthesis conditions and nanoparticle functionalization were shortened.

  1. Figure 4 contains too much information that one could barely understand the way of organization.

Response:

Figure 4 was re-arranged.

  1. The authors might need to shorten the subheadings (e.g. 3.1, 3.2).

Response:

Subheadings were changed according to the reviewer’s comments.

  1. When designing immunoassays, the authors coated that NPs with a gelatin layer. Nanozymes largely rely on their surface active sites to perform catalytic reactions. Does such modifications affect the nanozymatic activity of PBs?

Response:

We measured catalytic activity of coated and non-coated nanozymes. Gelatin coating decreased peroxidatic activity almost three-fold. These results were included in discussion (Section 3.4. and Figure S10).

Reviewer 3 Report

The manuscript “Prussian blue nanozymes with enhanced catalytic activity: size tuning and application in ELISA-like immunoassay” addresses an important nanozymes problem- process of preparation of ‘artificial peroxidase’ nanozymes with tunable properties. Authors demonstrated meticulous steps of optimization and characterizations of Prussian blue nanoparticles. They also demonstrate a practical utility of the assay, based on conjugated nanoparticles, for ELISA-like immunoassay. The work is interesting and precedent, although it is preliminary. I think it is acceptable after some revision, taking into account the following points.

Major points:

  1. Decimal point and decimal comma should not be used interchangeably.  A consistent decimal marker should be used throughout the paper. For example, in Table 2, both decimal point and  decimal comma are used. This should be fixed.
  2. The fitting equation for absorbance vs analyte concentration could be improved. Line 858. The author encountered “fitting issue”, which is manifest in low end of analyte concentration. It seems at low end, segmental linear regression was applied, whereas, at high end a smooth sigmoidal curve was applied. The correlation between absorbance and analyte concentration plays such an important role in determining analyte concentration of an unknown sample. As such, a better fitting equation could broaden the detected analyte concentration range but also improve the coefficient of determination. I suggest the authors perform a four-parameter logistic function analysis on absorbance vs analyte concentration. Please also include such fitting curve equation in the manuscript.
  3. Line 859. “true LOD value” is not a well-defined term. As mentioned in the point above, a four-parameter logistic function analysis would potentially yield a better

Minor points

  1. In Figure 2. Panel 25 mM. H2O2 concentration 5.5’s data is missing. Please acknowledge that.
  2. In Figure 7 C. It looks like the error bars are missing. Is it because n=1? If that is the case, Sample size is missing from data presentation for Figure 7 C.

Author Response

Reviewer #3

We express our great gratitude to the Reviewer for comments and thoughtful suggestions. Based on these comments and suggestions, we have made modifications to the original manuscript. In general, we made the manuscript more concise. Excessive information was removed or transferred to Supplementary Materials.

Key changes we made are summarized below:

  1. Last part of the Introduction was rewritten.
  2. Methods of nanoparticles characterization were moved to Supplementary Materials.
  3. Methodological details were removed from the Results and Discussion, except for Section 3.3 (first two paragraphs), where it is necessary for better understanding of the experiment design.
  4. TEM and SAED results for all samples were added (they can be found in the Supplementary Materials, figures S5-7). Data on specific surface area of nanoparticles was added (Fig 3E). Effect of protein coating on specific activity of nanozymes was studied (Section 3.4. and Figure S10).
  5. Curve fitting was performed using 4-parameter logistic model, according to the recommendations of the Referee. Details of curve fitting were added to the section 2.5.

All changes are highlighted with yellow.

We believe that the manuscript has been greatly improved and hope it has reached high journal's standard.

Major points:

  1. Decimal point and decimal comma should not be used interchangeably.  A consistent decimal marker should be used throughout the paper. For example, in Table 2, both decimal point and  decimal comma are used. This should be fixed.

Response:

Fixed.

  1. The fitting equation for absorbance vs analyte concentration could be improved. Line 858. The author encountered “fitting issue”, which is manifest in low end of analyte concentration. It seems at low end, segmental linear regression was applied, whereas, at high end a smooth sigmoidal curve was applied. The correlation between absorbance and analyte concentration plays such an important role in determining analyte concentration of an unknown sample. As such, a better fitting equation could broaden the detected analyte concentration range but also improve the coefficient of determination. I suggest the authors perform a four-parameter logistic function analysis on absorbance vs analyte concentration. Please also include such fitting curve equation in the manuscript.
  2. Line 859. “true LOD value” is not a well-defined term. As mentioned in the point above, a four-parameter logistic function analysis would potentially yield a better

Response:

Initially, we used 5-parameter logistic model for curve fitting, because it is recommended for non-symmetrical curves. When prepared revised version of manuscript, data were fitted to 4-parameter logistic regression model (using Origin2020b software), according to the Referee’s recommendation. Resulting plots can be found in Figure 6D,E. For PSA LOD changed from 0.064 to 0.189 ng/ml, for anti-tetanus IgG LOD remained unchanged (0.035 mIU/ml). Despite high coefficient of determination (more than 0.98 for both curves), lower part of regression curve remained segmented. We suppose that this is rather visual artifact, because Origin software is widely used for curve fitting and we have never had issues with it. Details of curve fitting are described in the Section 2.5. Section on results of immunoassay was re-written, incorrect terms were removed.

Minor points

  1. In Figure 2. Panel 25 mM. H2O2 concentration 5.5’s data is missing. Please acknowledge that.

Response:

The absence of a bar means an aggregation of nanoparticles. This explanation was added to the figure caption.

  1. In Figure 7 C. It looks like the error bars are missing. Is it because n=1? If that is the case, Sample size is missing from data presentation for Figure 7 C.

Response:

In this experiment, between-replicates reproducibility was high, therefore error bars are very short and are hidden by data points. n=3 for all experiments in this panel.